# Immunomodulatory fecal metabolites are associated with mortality in COVID-19 patients with respiratory failure

Matthew R. Stutz[1], Nicholas P. Dylla [2], Steven D. Pearson [1], Paola Lecompte-Osorio[1], Ravi Nayak[2], Maryam Khalid[2], Emerald Adler[2], Jaye Boissiere[2], Huaiying Lin[2], William Leiter [2], Jessica Little[2], Amber Rose [2], David Moran[2], Michael W. Mullowney [2], Krysta S. Wolfe [1], Christopher Lehmann [3], Matthew Odenwald[4], Mark De La Cruz[5], Mihai Giurcanu[6], Anne S. Pohlman[1], Jesse B. Hall[1], Jean-Luc Chaubard[2], Anitha Sundararajan[2], Ashley Sidebottom[2], John P. Kress[1], Eric G. Pamer [2,3] ✉ & Bhakti K. Patel [1] ✉

Respiratory failure and mortality from COVID-19 result from virus- and inflammation-induced lung tissue damage. The intestinal microbiome and associated metabolites are implicated in immune responses to respiratory viral infections, however their impact on progression of severe COVID-19 remains unclear. We prospectively enrolled 71 patients with COVID-19 associated critical illness, collected fecal specimens within 3 days of medical intensive care unit admission, defined microbiome compositions by shotgun metagenomic sequencing, and quantified microbiota-derived metabolites (NCT #04552834). Of the 71 patients, 39 survived and 32 died. Mortality was associated with increased representation of Proteobacteria in the fecal microbiota and decreased concentrations of fecal secondary bile acids and desaminotyrosine (DAT). A microbiome metabolic profile (MMP) that accounts for fecal secondary bile acids and desaminotyrosine concentrations was independently associated with progression of respiratory failure leading to mechanical ventilation. Our findings demonstrate that fecal microbiota composition and microbiota-derived metabolite concentrations can predict the trajectory of respiratory function and death in patients with severe SARS-Cov-2 infection and suggest that the gut-lung axis plays an important role in the recovery from COVID-19.

SARS-CoV-2, the cause of COVID-19, has infected nearly 500 million individuals, leading to over 6 million deaths worldwide as of April, 2022[1]. Mortality from infection can occur within days after symptoms develop or many weeks later. Respiratory failure and death during early infection are associated with high pulmonary SARS-CoV-2 titers,

epithelial and endothelial injury, infection of epithelial basal cells, neutrophil infiltration, and induction of interferon-stimulated genes (ISGs) while late mortality is associated with pulmonary infiltration with CD8 T cells expressing PD1, activated macrophages and reduced ISG transcript levels[2,3]. In the majority of COVID-19 patients, immune

mechanisms result in clearance of SARS-CoV-2 and resolution of inflammatory responses. Inflammatory cytokines, such as type I interferon, can reduce coronavirus loads during early pulmonary infection but can lead to increased lung pathology during later stages[4]. Dysfunctional immune responses, sometimes referred to as cytokine storm, can lead to progressive lung injury and death[5,6]. While the magnitude of early and late inflammatory responses to SARS-CoV-2 infection contribute to the wide range of clinical outcomes in COVID-19 patients, the underlying reasons for these disparities remain largely undefined[7].

Studies in mice demonstrated that the intestinal microbiome impacts pulmonary immune defenses against respiratory viral infections[8,9]. Commensal bacterial species produce metabolites that can activate systemic immune defenses and modulate inflammatory responses[10,11] and a diverse microbiome and its metabolic products can stimulate the host immune system and support immune homeostasis[12]. Production of bacterial metabolites, such as butyrate and secondary bile acids, can modulate development of inflammatory and regulatory T cell populations and provide defense against pathogens[13–17]. Microbiota-derived metabolites are increasingly recognized as contributing to respiratory antiviral defense[11]. For example, desaminotyrosine, a metabolite produced by a subset of intestinal microbes, has been shown in mice to enhance antiviral type I IFN responses and pulmonary clearance of influenza virus[18]. In patients undergoing hematopoietic cell transplantation (HCT), microbiota analyses revealed a fivefold increase in progression of viral respiratory tract infections among patients with reduced abundance of butyrate-producing commensal bacterial species[19]. The association between butyrate-producing bacteria and progression of respiratory viral illness was also seen in patients following renal transplantation[20].

Metagenomic sequencing studies have demonstrated that fecal microbiome compositions of COVID-19 patients are distinct from healthy subjects[21] and have revealed differences in microbiome composition and circulating markers of inflammation in patients with mild, moderate or severe respiratory disease[22–25]. Among patients with severe COVID-19, however, it remains unclear whether microbiome compositions and microbially derived, immunomodulatory metabolites are associated with progression of COVID-19-associated respiratory failure and mortality.

In this work, we profiled fecal microbiomes and targeted metabolites of patients admitted to the intensive care unit with COVID-19 and correlated these profiles with improvement or worsening of respiratory function and mortality. We find that reduced fecal concentrations of secondary bile acids and desaminotyrosine are associated with progression of respiratory failure and increased mortality in patients with severe COVID-19.

## Results

Between September 2020 and May 2021, 102 patients with COVID-19 were enrolled in a fecal collection protocol upon admission to the Medical Intensive Care Unit (ICU) at the University of Chicago Medical Center (Supplementary Fig. 1). This study was approved by the institutional review board (IRB 20-1102) at the University of Chicago and was registered on clinicaltrials.gove (NCT04552834). Of these, 71 patients produced a fecal sample within 72 h of enrollment (mean time of collection 24.7 h). Patient baseline information, clinical characteristics and antibiotic/antiviral treatment are stratified by mortality in Table 1. There were no significant differences between patients who survived versus died in terms of race, gender, diabetes, age, body mass index, hypertension, or chronic kidney disease. The two groups did not significantly differ in terms of treatment with antibiotics or COVID-19 specific therapies, such as steroids or remdesivir.

Microbiome compositions in each fecal sample, stratified by mortality, are shown in Fig. 1A and reveal higher densities of proteobacteria in patients who died of severe COVID-19 (Table 2). While

**Table 1 | Description of patient baseline demographics, past medical history, severity of illness and relevant medications**

| | Alive | Deceased | p |
|---|---|---|---|
| n | 39 | 32 | |
| **Baseline characteristics** | | | |
| Race (%) | | | 0.366 |
| Asian/Mideast Indian | 1 (2.6) | 0 (0.0) | |
| Black/African-American | 25 (64.1) | 19 (59.4) | |
| Hispanic | 2 (5.1) | 6 (18.8) | |
| More than one race | 1 (2.6) | 0 (0.0) | |
| Native Hawaiian/ Pacific Islander | 1 (2.6) | 0 (0.0) | |
| White | 9 (23.1) | 7 (21.9) | |
| Male (%) | 19 (48.7) | 18 (60.0) | 0.491 |
| Age (median [IQR]) | 58.97 [51.91, 69.01] | 66.21 [56.56, 72.66] | 0.168 |
| Body mass index (median [IQR]) | 31.51 [27.96, 38.78] | 30.73 [26.83, 34.08] | 0.432 |
| Hypertension (%) | 23 (59.0) | 23 (71.9) | 0.377 |
| Hyperlipidemia (%) | 12 (30.8) | 11 (34.4) | 0.946 |
| Diabetes (%) | 13 (33.3) | 12 (37.5) | 0.908 |
| Cancer (%) | 6 (15.4) | 3 (9.4) | 0.69 |
| Chronic kidney disease (%) | 3 (7.7) | 8 (25.0) | 0.094 |
| **Clinical characteristics** | | | |
| Charlson comorbidity index (median [IQR]) | 3.00 [2.00, 4.50] | 4.00 [2.00, 5.00] | 0.551 |
| SOFA score (median [IQR]) | 5.00 [4.00, 9.00] | 9.00 [8.00, 9.00] | <0.001 |
| APACHE score (median [IQR]) | 18.00 [13.00, 22.50] | 23.00 [19.00, 30.25] | 0.002 |
| Days from symptom onset (median [IQR]) | 5.00 [3.00, 7.00] | 5.00 [2.75, 8.00] | 0.912 |
| Admission (%) | | | 0.229 |
| Emergency department | 25 (64.1) | 14 (43.8) | |
| Hospital medicine | 8 (20.5) | 10 (31.2) | |
| Outside hospital | 6 (15.4) | 8 (25.0) | |
| **Medication administration** | | | |
| Antivirals (%) | | | |
| Remdesivir treatment | 28 (71.8) | 22 (68.8) | 0.985 |
| Steroid treatment | 28 (71.8) | 23 (71.9) | 1 |
| Antibiotics (%) | | | |
| Betalactams | 8 (20.5) | 5 (15.6) | 0.825 |
| Levofloxicin | 1 (2.6) | 0 (0.0) | 1 |
| Vancomycin | 9 (23.1) | 13 (40.6) | 0.183 |
| Metronidazole | 3 (7.7) | 4 (12.5) | 0.782 |
| Macrolides | 8 (20.5) | 6 (18.8) | 1 |
| Doxycycline | 5 (12.8) | 1 (3.1) | 0.302 |
| Trimethoprim-Sulfamethoxazole | 5 (12.8) | 2 (6.2) | 0.6 |
| Aminoglycosides | 1 (2.6) | 0 (0.0) | 1 |

Adequate treatment with COVID-19 specific therapy included at least three consecutive days of therapy during index hospitalization. Sufficient total dose of steroids was the equivalent of 18 mg of dexamethasone and 400 mg of Remdesivir. Only antibiotics received 72 h prior to fecal specimen collection are represented (n = 71 independent samples from patients). Sequential Organ Failure Assessment (SOFA) and Acute Physiology and Chronic Health Evaluation (APACHE) assess ICU clinical status. Categorical variables were compared using ta two-tailed, chi-squared test, while continuous variables were compared using the Wilcoxon rank-sum, two-tailed. Unadjusted p-values are presented as exact values.

microbiome alpha diversity and species richness and evenness did not differ between patients who survived versus those who died (Inverse Simpson: 12.8 vs 13.2, $W(71) = 628$, $p = 0.968$, two-tailed test; Shannon Index: 3.09 vs 2.89, $W(71) = 650$, $p = 0.77$, two-tailed test; Species Richness: 157 vs 148, $W(71) = 654$, $p = 0.737$, two-tailed test; Species Evenness: 0.61 vs 0.58, $W(71) = 637$, $p = 0.886$, two-tailed test) (Fig. 1B,

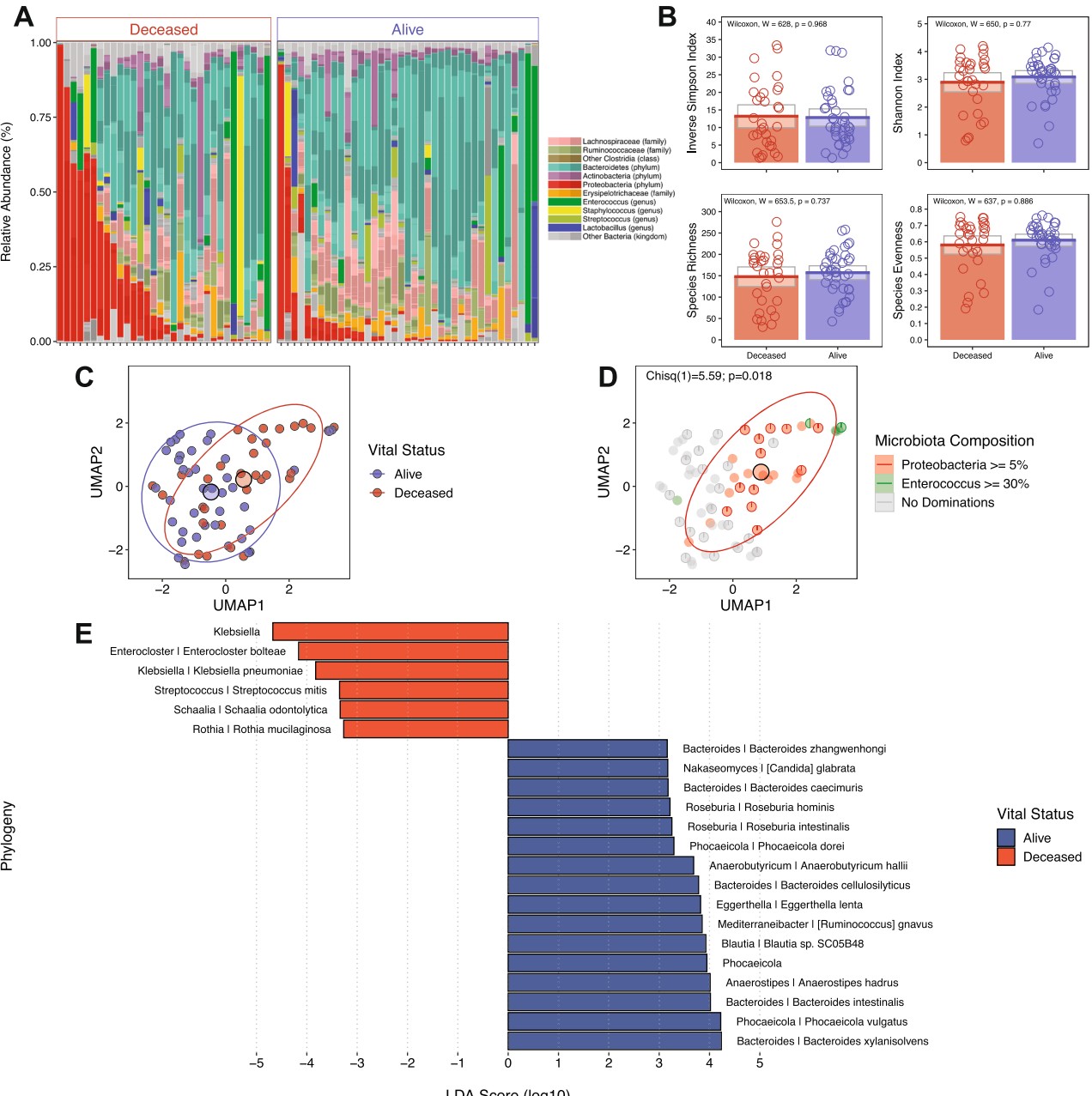

**Fig. 1 | Fecal microbiome composition in patients with severe COVID-19 stratified by mortality. A** Shotgun metagenomics-based taxonomy plots stratified by survival where taxa are shaded to biologically relevant levels (legend to the right). **B** Alpha diversity (Inverse Simpson and Shannon Index) plots and Species Richness and Evenness stratified by survival where colored bars represent the average value for survival (blue bars: alive, red bars: deceased) while gray boxes denote 95% confidence intervals. Wilcoxon rank-sum, two-tailed tests were implemented and *p*-values were adjusted via the Benjamini-Hochberg method. **C** Uniform Manifold Approximation and Projection (UMAP) from shotgun metagenomics-based taxonomy, colored by survival (blue points: alive and red points: deceased) with centroids and 95% CI ellipses. **D** UMAP colored by expansion of *Enterococcus* (green), *Proteobacteria* (red), both *Enterococcus* and *Proteobacteria* (red and green halves), and no expansions (gray) with centroids and 95% CI ellipse. A two-tailed, chi-squared test was used to compare expansions to vital outcomes. **E** A Linear discriminant analysis effect size (LEfSe) showing the significant (Wilcoxon rank-sum, two-tailed, *p* ≤ 0.05) effect sizes of taxa between survival groups (blue bars: alive, red bars: deceased). A linear discriminant analysis was performed in lieu of adjusting for multiple comparisons. *n* = 71 independent samples from patients.

Table 2), dimension reduction of microbiome compositions by Uniform Manifold Approximation and Projection (UMAP) demonstrated distinct clustering of patients who died versus patients who recovered from COVID-19 (Fig. 1C), with statistically significant overlap of Proteobacteria dominated clusters with mortality (Fig. 1D, χ²(1, 71) = 5.59, *p* = 0.018). Linear discriminant analysis effect size (LEfSe) using shotgun metagenomics-based taxonomy data indicated survival was associated with increased representation of obligate anaerobic bacterial species belonging to the *Bacteroidaceae* and *Lachnospiraceae*

families while mortality was associated with expansion of *Enterobacteriaceae* (Fig. 1E). Furthermore, *Proteobacteria* expansion (relative abundance of >5%) was significantly associated with higher mortality (Table 2, *W*(71) = 456, *p* = 0.035, two-tailed test).

Although microbiota compositions differed between COVID-19 patients who survived versus died of COVID-19, comparison of KEGG metabolic pathways did not identify significant differences between the two groups (Supplementary Fig. 2). Because most metabolic pathways included in this comparison contribute to general bacterial

**Table 2 | Characteristics of the fecal microbiome stratified by mortality**

| | Alive | Deceased | p |
|---|---|---|---|
| n | 39 | 32 | |
| **Microbiological characteristics** | | | |
| Diversity: (median [IQR]) | | | |
| Inverse Simpson | 10.77 [7.53, 18.05] | 10.80 [5.74, 20.70] | 0.931 |
| Relative abundance: (% of group) | | | |
| Proteobacteria domination | 9 (23.1) | 16 (50.0) | 0.035 |
| Enterococcus domination | 3 (7.7) | 3 (9.4) | 1 |
| Metabolites: (median [IQR]) | | | |
| Desaminotyrosine (µM) | 28.10 [22.74, 46.94] | 22.63 [21.00, 32.30] | 0.028 |
| Lithocholic acid (µg/mL) | 61.61 [4.30, 252.61] | 9.76 [0.24, 60.07] | 0.008 |
| Deoxycholic acid (µg/mL) | 74.09 [18.61, 246.34] | 10.68 [0.49, 62.50] | 0.003 |
| Isodeoxycholic acid (µg/mL) | 7.60 [0.70, 25.09] | 2.89 [0.06, 10.93] | 0.038 |
| Cholic acid (µg/mL) | 16.21 [1.48, 215.59] | 7.92 [0.79, 88.15] | 0.214 |
| Glycocholic acid (µg/mL) | 0.43 [0.18, 2.83] | 0.45 [0.03, 11.78] | 0.511 |
| 3-oxolithocholic acid (µg/mL) | 4.71 [0.64, 54.90] | 1.88 [0.06, 14.03] | 0.086 |
| Alloisolithocholic acid (µg/mL | 0.44 [0.00, 2.79] | 0.20 [0.00, 6.18] | 0.482 |
| Taurocholic acid (µg/mL) | 0.55 [0.22, 2.23] | 0.30 [0.08, 24.06] | 0.391 |
| Butyrate (mM) | 0.51 [0.20, 2.39] | 0.29 [0.06, 0.97] | 0.286 |
| Propionate (mM) | 1.55 [0.47, 4.94] | 0.75 [0.19, 1.62] | 0.169 |
| Acetate (mM) | 3.87 [2.29, 20.44] | 3.13 [1.17, 11.23] | 0.175 |
| Succinate (mM) | 0.59 [0.26, 1.58] | 0.50 [0.18, 1.38] | 0.431 |
| Microbiome metabolic profile (median [IQR]) | 1.00 [0.00, 2.00] | 2.00 [1.00–4.00] | <0.001 |

Categorical variables were compared using a two-tailed, chi-squared test, while continuous variables were compared using the Wilcoxon rank-sum, two-tailed test ($n = 71$ independent samples from patients). Unadjusted p-values are presented as exact values.

physiology, and thus might conceal less prevalent pathways that contribute to COVID-19 pathogenesis, we next focused on genes encoding bacterial antibiotic resistance, bacteriocins and toxins/hemolysins/cytolysins (Fig. 2A–C). After correcting for multiple comparisons, we did not detect statistically significant differences between the two groups. However, there was a trend towards increased representation of antibiotic-resistance genes in the patient group that died of COVID-19. Because secondary bile acids and butyrate have been shown to be immunomodulatory, we specifically quantified the frequencies of genes encoding 3βHydroxysteroid dehydrogenase (3βHSDH), 5αReductase (5AR), the Bai operon and Butyrate Kinase but did not detect significant differences between the two patient groups (Fig. 2D).

Although the metabolic output of the intestinal microbiota is determined by its composition, other factors, such as diet and the host's state of immune activation and inflammation, also impact metabolite production. Thus, compositionally similar microbiota can establish distinct fecal metabolomes, resulting from changes in diet, medications and/or the host's state of immune activation[25]. To determine whether fecal metabolomes differed between patients who survived or died from COVID-19, we performed GC- and LC-MS on fecal samples to quantify a range of fatty acids, amino acids, bile acids and other metabolites known to be produced by commensal bacteria. Differences in representation of 92 metabolites are presented in a heat map of normalized values and are expressed as fold changes relative to the mean value for all samples (Supplementary Fig. 3). A volcano plot more clearly identifies metabolites associated with survival, including secondary bile acids, indole-3-carboxaldehyde and desaminotyrosine (Fig. 3A). Deoxycholic acid, lithocholic acid, isodeoxycholic acid and desaminotyrosine were each associated with survival (Fig. 3A, B, Supplementary Fig. 3). Univariate analyses demonstrated significant associations between lithocholate ($W(67) = 765$, $p = 0.021$, two-tailed test), deoxycholate ($W(67) = 791$, $p = 0.015$, two-tailed test), isodeoxycholate ($W(67) = 720$, $p = 0.048$, two-tailed test), and

desaminotyrosine ($W(63) = 653$, $p = 0.046$, two-tailed test) (Fig. 3B, Table 2) and survival of COVID-19. Butyrate, acetate and propionate concentrations, while reduced in COVID-19 patients who died, did not achieve significance (Supplementary Fig. 3, Table 2).

Given the parallel and mechanistically distinct contributions of intestinal microbes and their metabolites to immune modulation and inflammatory responses, we developed the Microbiome Metabolite Profile (MMP) to more comprehensively associate the microbiome's function with clinical outcomes. The components of the MMP, deoxycholic acid, lithocholic acid, isodeoxycholic acid, and desaminotyrosine, were selected based on association with survival and their plausible immune regulatory and antiviral roles during SARS-CoV-2 infection (Table 3). With respect to mortality, the MMP demonstrated an AUC = 0.74 ([CI]: 0.628–0.860) with negative predictive value of 0.67 and positive predictive value of 0.75 (Fig. 4A). Kaplan–Meier survival curves demonstrate 33.5% mortality in the low MMP (MMP = 0–1) group compared to 89.3% mortality in high MMP (MMP = 2–4) group ($n = 68$, $p = 0.0024$) (Fig. 4B). To test for independent association with mortality, the MMP, as the sole evaluator of microbiome health, was included in a Cox proportional hazard model along with other variables with univariable $p$-value < 0.3. This model demonstrated that at any point in the study, patients with a high MMP score were 65% ([CI]:18–231%) more likely to die than patients with a low MMP score ([HR]:1.65, [CI]:1.18–2.31, $p = 0.003$) (Table 4).

Although the 71 patients admitted to the ICU had severe respiratory compromise, a subset of 50 patients did not initially require mechanical ventilation and were treated with high-flow oxygen by nasal canula (HFNC) (Fig. 5). The course of respiratory failure in this group included 20 patients who progressed from HFNC to endotracheal intubation and 30 patients who de-escalated to low flow nasal cannula (LFNC). Patients in whom transitions could not be identified were most frequently already intubated at ICU admission or did not require respiratory support other than LFNC. Patient and microbiologic characteristics are stratified by progression of respiratory

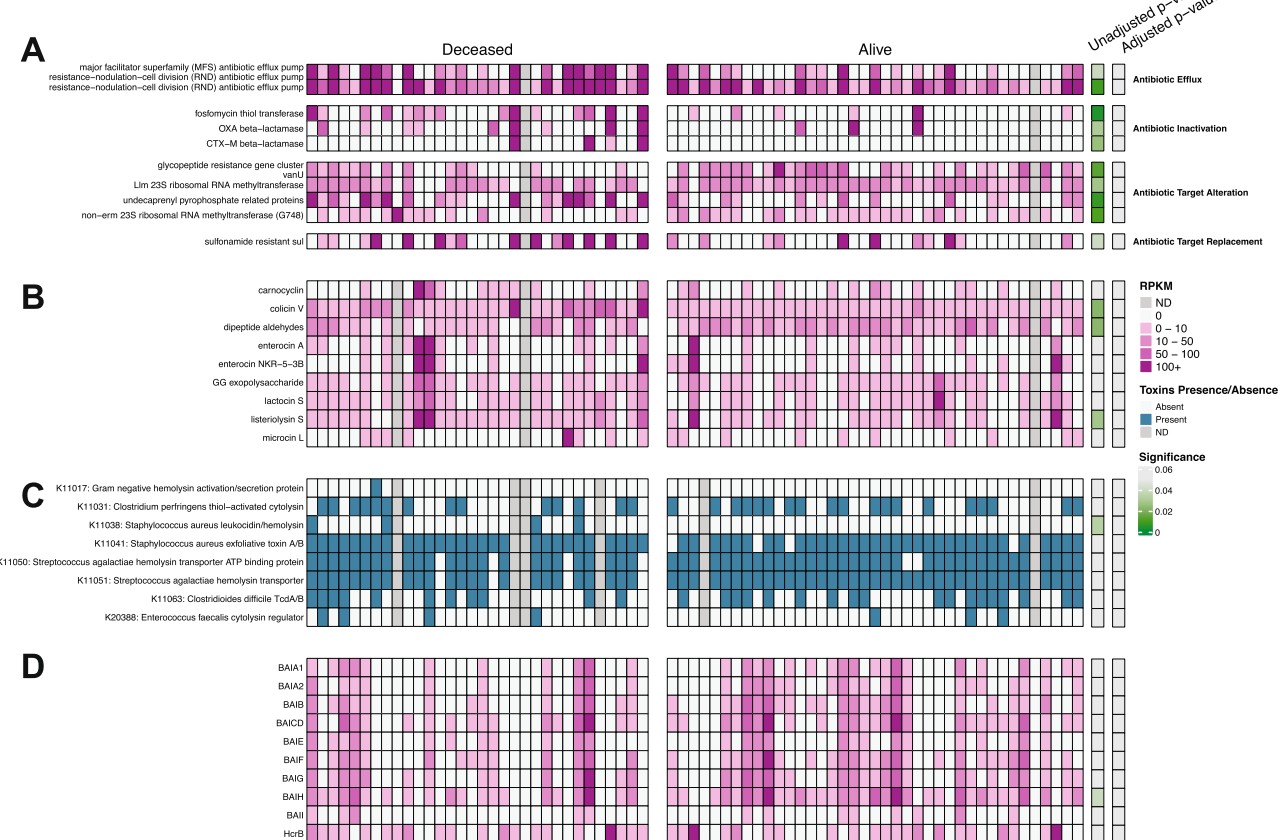

**Fig. 2 | Representation of genes encoding antibiotic resistance, toxins and metabolite production stratified by Mortality.** Panel **A** displays genes encoding for antibiotic resistance. Panel **B** displays genes encoding for bacteriocins. Panel **C** displays genes encoding for toxins/hemolysins/cytolysins. Panel **D** displays the genes responsible for bile acid conversion, as well as butyrate-related enzymes and desaminotyrosine. Genes in Panels **A**, **B**, **D** were quantified using RPKM values (shades of pink) while toxin genes in Panel **C** were determined as presence/absence (blue/white). Gray boxes show missing data. *P*-values and adjusted *p*-values (via Benjamini-Hochberg method) were obtained from Wilcoxon rank-sum, two-tailed tests (**A**, **B**, **D**) and a two-tailed, chi-squared test (**C**) and are shown as shading from non-significant (gray) to statistically significant (green). n = 71 independent samples from patients.

failure in Supplementary Table 1. Variables found to be significant on univariable analysis were included in a multivariable logistic regression model which demonstrated higher MMP to be independently associated with progression of respiratory failure requiring intubation ([HR]: 1.11; [CI]:1.02–1.20; *p* = 0.025) (Table 5).

## Discussion

The high mortality associated with COVID-19 results from viral injury to lung tissue, overly robust inflammatory responses, and subsequent alveolar damage. While the relative contributions of these factors vary from one patient to another, autopsy studies suggest that in fatal COVID-19, all three play a role[2,3]. We used a multifaceted approach to assess the integrity of the fecal microbiome during COVID-19 associated critical illness and found that the intestinal microbiome and its metabolites at the time of ICU admission are independently associated with the need for intubation and survival. Microbiota-derived metabolites potentially contribute to viral clearance, modulation of inflammation and reestablishment of epithelial integrity. While conventional assessments of microbial α-diversity did not distinguish survivors from patients who died, in multivariable models, a subset of microbially derived fecal metabolites correlated with survival. Furthermore, the associations found in this study persisted after taking antibiotic treatment, duration of symptoms, and severity of illness into account.

While previous studies have correlated increased frequencies of Proteobacteria and reduced frequencies of obligate anaerobic commensal species with SARS-CoV-2 infection[5,22,24], our study provides the first correlation with COVID-19 mortality and need for intubation.

Experimental SARS-CoV-2 infection of rhesus macaques resulted in expansion of Proteobacteria during peak infection[26], suggesting that the microbiota compositional changes we have detected in our patients may result from the systemic viral infection. Whether the expansion of Proteobacteria in the gut microbiota contributes to respiratory disease progression requires further study.

Although our study is limited to associating microbiome and metabolic features with progression of respiratory failure and mortality, there are plausible mechanisms by which the identified metabolites might reduce mortality in patients with COVID-19. Immune responses during early stages of viral infection and regulation of lung inflammation at later stages are likely impacted by the microbiome and its metabolites. Our finding that increased concentrations of fecal secondary bile acids are associated with improved outcomes may result from their recently described impact on differentiation of CD4 Th17 and Treg cells[16,17,27]. The secondary bile acid deoxycholate is produced by a subset of commensal bacteria[28] and is further converted by bacterial species expressing 3αHSDH and 3βHSDH to isodeoxycholate, which is less toxic to mammalian cells and commensal bacterial species[29]. Isodeoxycholate also renders dendritic cells less immunostimulatory, thereby enhancing generation of peripherally induced T regulatory cells[11,16]. Some bacterial strains belonging to the Bacteroidales order express 5AR and 3β-HSDH, enabling them to generate alloisolithocholate from bile acid intermediates along the Bai pathway[30]. Importantly, alloisolithocholate can inhibit gram positive pathogens and also enhance the development of T regulatory cells[17,27,30]. Although a recent study demonstrated increased T

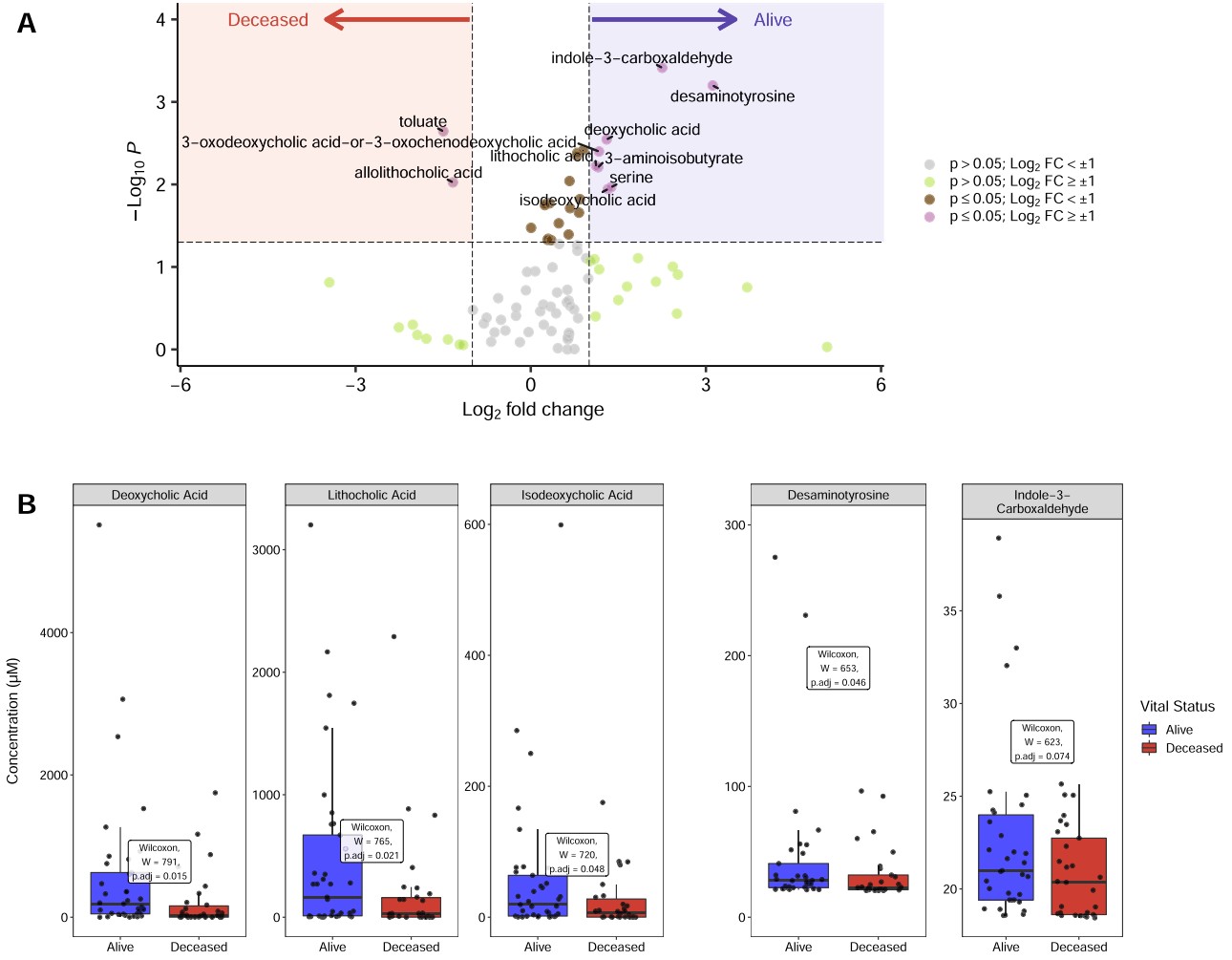

**Fig. 3 | Qualitative and quantitative fecal metabolomic analyses. A** Volcano plot of normalized metabolite concentrations, where values above the horizontal line (Wilcoxon rank-sum, two-tailed, unadjusted $p$-value > 0.05) and log2 fold-change values >= 1 were used to identify metabolites associated with survival. Red shading shows compounds more abundant in the deceased population while blue shading displays compounds that were more abundant in the alive population. Gray points denote $p$-values > 0.05 and log2 fold-change values < ±1; green points denote $p$-values > 0.05 and log2 fold-change values >± 1; brown points denote $p$-values < 0.05 and log2 fold-change values < ±1; and purple points denote $p$-values < 0.05 and log2 fold-change values > ±1. **B** Metabolites identified in panel **A** for survival groups (blue: alive and red: deceased) were subsequently quantified in fecal extracts by LC-MS and are shown as boxplots and compared using Wilcoxon rank-sum, two-tailed tests with $p$-values adjusted for multiple comparisons via the Benjamini-Hochberg method ($n = 68$ independent samples from patients). Bile acids, desaminotyrosine and indole-3-carboxaldehyde are in units of μM. Boxes show interquartile ranges (IQR) where the center black line represents the median and the whiskers (vertical black lines) extend to $1.5 × IQR$ or to the minimum and maximum value, whichever is closest to the median.

regulatory cell frequencies in the bloodstream of patients with severe COVID-19, it remains unclear whether they contribute to or ameliorate pulmonary pathology[31].

Our finding that the frequency of genes encoding enzymes that mediate secondary bile acid synthesis did not differ between patients

## Table 3 | Quantitated metabolomic compounds (μM) and threshold values developed for the Microbiome Metabolite Profile

| Compound | Points | |
|---|---|---|
| | 0 | 1 |
| Deoxycholic acid | ≥89.92 (μM) | <89.92 (μM) |
| Isodeoxycholic acid | ≥0.97 (μM) | <0.97 (μM) |
| Lithocholic acid | ≥258.25 (μM) | <258.25 (μM) |
| Desaminotyrosine | ≥21.31 (μM) | <21.31 (μM) |

Variables chosen based on biologic plausibility and statistical significance. Higher scores indicated microbiome dysfunction.

with respect to mortality while fecal secondary bile acid concentrations did (Fig. 3), suggests that metabolic profiling represents a more sensitive measure of the potential impact of the microbiome on host physiology and immune defense than metagenomic sequencing. It also demonstrates that metabolite production is not only dependent on the presence of genes but host factors such as diet, inflammation, and the presence of essential co-factors. In addition, a functional assessment of the integrity of the microbiome through metabolites is advantageous clinically as they can be rapidly quantified.

Type I interferon responses to SARS-CoV-2 infection contribute to viral clearance during early stages of infection and reduced type I interferon levels have been associated with more severe COVID-19[32,33]. Previous studies in mice have demonstrated that the microbially derived metabolite desaminotyrosine (DAT), also known as 4-hydroxyphenylpropionic acid, amplifies type I interferon production during early stages of influenza virus infection, thereby enhancing host resistance and reducing lung injury[18]. DAT, first described as a product of flavin degradation[34], is also a product of tyrosine metabolism by a subset of intestinal microbes such as *Clostridium sporogenes*[35]. Our

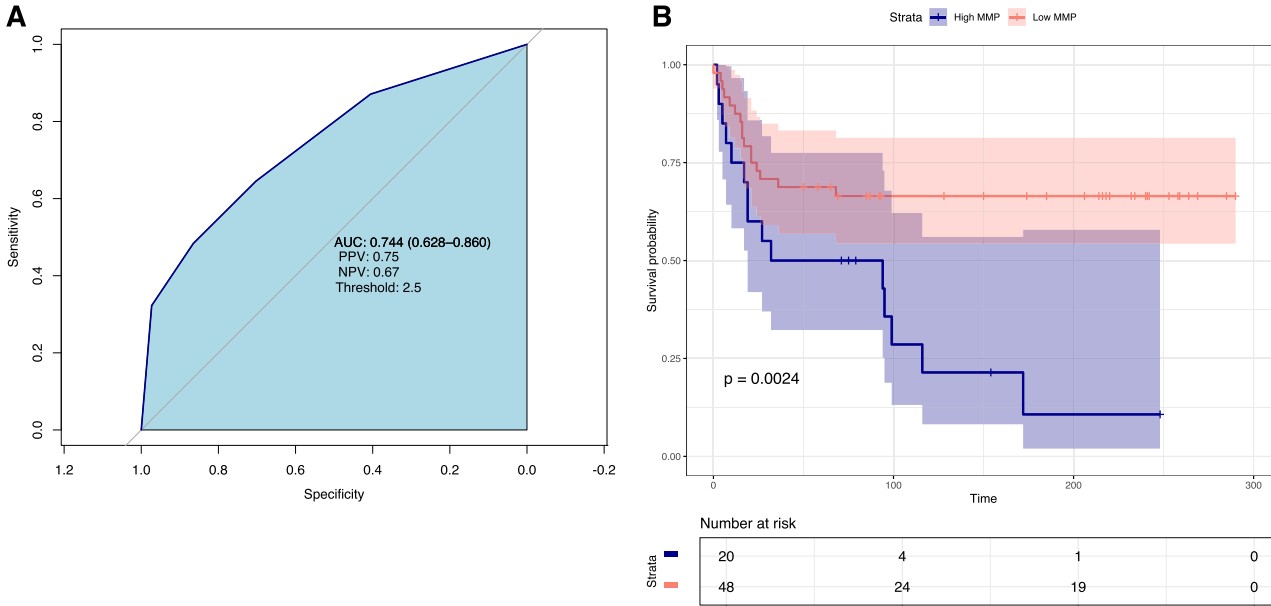

**Fig. 4 | A Microbiome Metabolite Profile (MMP) predicts mortality in patients with severe COVID-19. A** Area under the curve (AUC) for the microbiome metabolite profile and mortality. AUC = 0.744. Positive predictive value (PPV) = 0.75 and negative predictive value (NPV) = 0.67. **B** Kaplan–Meier survival curves stratified by low MMP scores (0–1, red shading) versus high MMP score (2–4, blue shading) are plotted. Time in days is presented on the *X*-axis. Log-rank test was used to assess significant differences. *n* = 68 independent samples from patients.

finding that increased concentrations of DAT in fecal samples is associated with recovery from COVID-19 suggests that microbiota-mediated modulation of type I interferon signaling attenuates lung injury in patients with severe SARS-Cov-2 infection.

Our study has several limitations. First, because our study includes patients treated at a single institution, our findings may not readily extend to patients receiving treatment in other medical centers. Second, although we have demonstrated significant associations between the progression of respiratory failure and the paucity of potentially immunomodulatory metabolites, a larger sample size might have identified additional important correlations. Third, our patients were enrolled in this study during a period when the treatment of COVID-19 was evolving and thus the impact of microbiota-derived metabolites on the course of respiratory failure may have evolved over time.

### Table 4 | Cox proportional hazards regression model for mortality

| Characteristic | HR | 95% CI | *p*-value |
|---|---|---|---|
| Age | 1.05 | 1.01, 1.08 | 0.010 |
| Chronic kidney disease | 1.09 | 0.42, 2.81 | 0.861 |
| SOFA score | 1.44 | 1.23, 1.68 | 3.59e-06 |
| Vancomycin administration | 0.41 | 0.17, 1.00 | 0.051 |
| Admission location | | | |
| Emergency department | – | – | |
| Hospital medicine | 1.49 | 0.59, 3.73 | 0.399 |
| Outside hospital | 0.18 | 0.04, 0.86 | 0.032 |
| Microbiome metabolic profile | 1.65 | 1.18, 2.31 | 0.003 |

Variables with unadjusted *p*-values < 0.3 from the univariate analysis (Table 2) were included in the multivariate analysis. Unadjusted *p*-values were obtained from a likelihood ratio test and reported as exact values (*n* = 68 independent samples from patients).
*HR* hazard ratio, *CI* confidence interval.

We demonstrate that microbiome composition and a subset of microbiota-derived metabolites are independently associated with survival and the trajectory of respiratory failure among patients admitted to the ICU with COVID-19. Given the malleability of the microbiome's composition and function, identification and characterization of metabolites associated with improved clinical outcomes may enable therapeutic interventions that include microbiome manipulation and augmentation.

## Methods

### Study design and patient enrollment

This was a prospective observational cohort study which took place at a single urban academic medical center in the United States. This study was approved by the University of Chicago Institutional Review Board and has been registered at clinical-trials.gov as NCT #04552834. Patients with COVID-19-associated respiratory failure or shock admitted to the medical ICU were included in the study. For the purposes of the inclusion criteria, respiratory failure was defined by the receipt of non-invasive positive pressure ventilation (NIPPV), high flow nasal cannula (HFNC), or invasive mechanical ventilation. Shock was defined by the receipt of vasoactive medications. Exclusion criteria included: age <18 years, pregnancy, and prior cardiac arrest during admission of interest. COVID-19 diagnosis was confirmed by reverse transcriptase-polymerase chain reaction of nasal pharyngeal swabs. This project received institutional review board (IRB) approval from the University of Chicago (20-1102). Informed consent was obtained from the patient or surrogate decision makers prior to enrollment. Enrollment began in September 2020 and concluded in May of 2021. Patient enrollment is described in Supplement Fig. 1. Patients were followed up to 1 year following study completion by chart review and telephone.

### Specimen collection and analysis

Fecal samples were collected as soon as possible following ICU admission, immediately refrigerated and aliquoted and frozen at −80 C

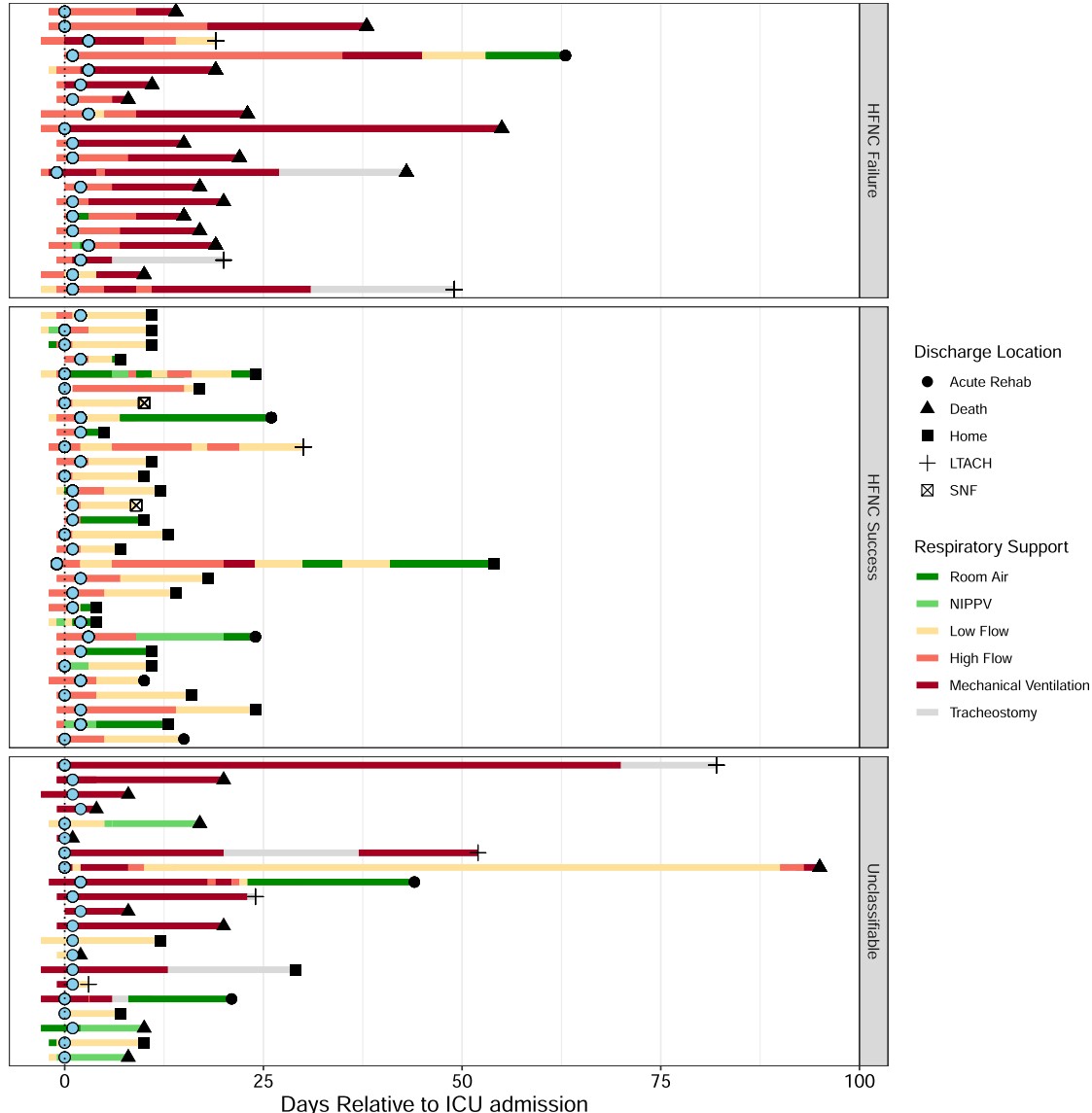

**Fig. 5 | Progression of Respiratory Failure Stratified by Trajectory.** Each row represents an individual patient course. Blue dots represent initial fecal samples collected within 3 days of ICU admission. Figure is stratified by patients who transitioned from high flow nasal cannula to low flow nasal cannula versus those who progressed to endotracheal intubation and received mechanical ventilation. Patients in whom a transition could not be identified were labeled unclassifiable. Shapes are denoted as the discharge location while colored bars denote the type of respiratory support. LTACH long term acute care hospital, SNF skilled nursing facility.

within 24 h. Samples which were collected within the first 72 h of ICU hospitalization at University of Chicago were included in this analysis. This time frame was chosen to focus the study on whether the fecal microbiota in the early course of illness influences clinical trajectories and outcomes.

## Metagenomic analyses

Fecal samples underwent metagenomic shotgun DNA sequencing. Samples underwent mechanical disruptions with a bead beater (BioSpec Product) and were further purified with QIAamp mini spin columns (Qiagen). Purified DNA was quantified with a Qubit 2.0 fluorometer and sequenced on the Illumina HiSeq platform. Fecal samples from clinical patients were prepared in batches of 60–72 in paired-end (PE) libraries with insert size around 350 bp for each sample. High-throughput sequencing on Illumina NextSeq 500/NovaSeq 6000 produced around 7 to 8 million PE reads per sample with read length of 149/159 bp. Adapters were trimmed off from the raw reads, and their quality was assessed and controlled using

Trimmomatic (v.0.39)[36]. Reads mapped to the human genome were be identified and removed by kneaddata (v0.7.10). Microbial reads were assembled using MEGAHIT (v1.2.9)[37] and genes were called by prodigal and annotated with prokka (v.1.14.6)[38]. The translated proteins for each fecal microbiome were functionally profiled by eggnog mapper (2.0.1b)[39] against their default precomputed orthologous groups and phylogenies from the EggNOG database including presence/absence of KOs (KEGG Orthologies) from the KEGG database[40]. Genes encoding for antibiotic resistance[41] and bacteriocins were queried against high-quality shotgun reads using DIAMOND (v2.0.4.) with a filter threshold of ≥80% identity and ≥80% protein coverage. Reads are reported in reads per million normalized by the length of the gene, in kilobases (RPKM). Toxins/hemolysins/cytolysins genes were obtained from the EggNOGG database alignment and returned as presence/absence of a KO. Taxonomy was profiled using Kraken2 on PATRIC v3.5.0. Alpha diversity was determined by Inverse Simpson and Shannon Index and species richness and evenness were also determined.

**Table 5 | Multivariable regression model for high flow nasal cannula failure**

| Characteristic | HR | 95% CI | *p*-value |
|---|---|---|---|
| Age | 1.00 | 1.00, 1.01 | 0.527 |
| Chronic kidney disease | 0.96 | 0.68, 1.34 | 0.801 |
| Steroid treatment | 1.11 | 0.87, 1.42 | 0.424 |
| SOFA score | 1.12 | 1.07, 1.17 | 1.12e-05 |
| Doxycycline | 0.77 | 0.55, 1.07 | 0.125 |
| Microbiome metabolic profile | 1.11 | 1.02, 1.20 | 0.025 |

Variables with *p*-values < 0.3 from univariable were included in the multivariable analysis. Unadjusted *p*-values were obtained from a likelihood ratio test and reported as exact values (*n* = 68 independent samples from patients).
*HR* hazard ratio, *CI* confidence interval.

## Metabolomic analyses

Three short chain fatty acids (butyrate, acetate, and propionate) and succinate were derivatized with pentafluorobenzyl bromide (PFBBr) and analyzed via negative ion collision induced-gas chromatography-mass spectrometry ([−]CI-GC-MS, Agilent 8890). Eight bile acids (primary: cholic acid; conjugated primary: glycocholic acid, taurocholic acid; secondary: deoxycholic acid, lithocholic acid, isodeoxycholic acid, alloisolithocholic acid and 3-oxolithocholic acid) (μg/mL) were quantified by negative mode liquid chromatography-electrospray ionization-quadrupole time-of-flight-MS ([−]LC-ESI-QTOF-MS, Agilent 6546). Desaminotyrosine and indole-3-carboxaldehyde were analyzed via UPLC-QqQ LC-MS (μM). Ninety-two total additional compounds were relatively quantified using normalized peak areas relative to internal standards. Additional details regarding metabolite analysis can be found in Supplementary Methods. Insufficient sample precluded metabolite analysis of seven patients. All metabolomic data was submitted to the EMBL-EBI MetaboLights database (DOI: 10.1093/nar/gkz1019, PMID:31691833) with the identifier MTBLS5288 and accession number MTBLS5288. The complete dataset can be accessed at https://www.ebi.ac.uk/metabolights/MTBLS5288.

## Development of the microbiome metabolite profile

The microbiome metabolite profile (MMP), was developed as an aggregate of selected metabolic features of the microbiome that might reflect its functional potential using the R programming language (v 4.1.1). Metabolites were down-selected using both the results from the volcano analysis (Fig. 3A) as well as metabolites that maintained biological plausibility (i.e. deoxycholic acid, isodeoxycholic acid, lithocholic acid, and desaminotyrosine). Optimized thresholds for these four compounds (concentrations) were determined using the Youden Index (cutpointr:cutpointr, v1.1.1) which individually selects the metabolite concentration that optimizes sensitivity and specificity, where outcomes were binary (alive/deceased; Table 3)[42]. If concentrations were less than the optimized thresholds, one point was assigned to that compound, where scores that correlate to survival were zero and scores that correlate to death were one; a minimum of 0 points and a maximum of 4 points could be assigned. To assess the model, a receiver operator characteristic curve (ROC) analysis was performed (Fig. 4A). In addition to the AUC score, the optimized threshold, positive predicted value (PPV) and negative predicted value (NPV) were also calculated.

## Clinical data

Clinical data was obtained through a data extraction procedure of the electronic medical record, confirmed by manual chart review or by manual chart review alone, and stored in a REDCap secure online database (version LTS 11.1.7).

## Statistical analysis

All statistical analyses were conducted using the R programming language (version 4.1.1). Adjusted *p*-values of the tests were considered to be statistically significant for all analyses conducted if $p \leq 0.05$. In some instances, unadjusted p-values were also displayed. Continuous variables were compared between the survival groups using Wilcoxon rank-sum test (rstatix::wilcox_test) and p-values were adjusted following the Benjamini-Hochberg method (rstatix:: adjust_pvalue). Categorial variables were compared using the chi-squared test (chisq.test). Kaplan–Meier curves for survival endpoints were generated as well as stratified by selected risk factors such as protobacteria abundance (normal/abnormal; survival::Surv, survfit, ggsurvplot) and a log-rank test was used to assess significant differences (glm). A Cox proportional hazards regression model for mortality and a relative risk regression model for progression of respiratory failure indicator were used to estimate the effects of microbiome and metabolites adjusting for known risk factors (survival::coxph).

## Reporting summary

Further information on research design is available in the Nature Research Reporting Summary linked to this article.

## Data availability

All data are available to the public. Raw sequencing data generated in this study have been deposited onto the National Center for Biotechnology Information (NCBI) Sequence Read Archive (SRA) under accession number PRJNA842425. Raw metabolomic data generated in this study have been deposited onto Metabolights under accession number MTBLS5288. All processed data in this study are hosted on both GitHub (https://github.com/DFI-Bioinformatics/SARS-CoV-2) as well as Zenodo (https://doi.org/10.5281/zenodo.6858446).

## Code availability

All code used in this study for analyses and to generate figures is available both at GitHub (https://github.com/DFI-Bioinformatics/SARS-CoV-2) and Zenodo (https://doi.org/10.5281/zenodo.6858446).

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

## Acknowledgements

The authors acknowledge the important contributions of the late Eric Littmann to the work contained in this manuscript. Funding: NIH T32 HL-007605 (M.R.S. and & S.D.P.), NIH P01 CA023766 and Duchossois Family Institute (E.G.P.), NIH/NHLBI K23 HL148387 (B.K.P.).

## Author contributions

M.R.S., J.P.K., E.G.P., and B.K.P. conceived and designed the study. N.P.D. and H.L. performed data analysis and interpretation and drafted the figures and tables. R.N., M.K., E.A., and J.B. drafted the clinical study protocol, consented patients and facilitated collection of clinical samples. W.L., J.L., A.R., D.M., M.W.M., J.-L.C., and A. Sidebottom performed and analyzed metabolomic data. S.D.P., P.L.-O., K.S.W., C.L., M.O., M.D.L.C., A.S.P., and J.B.H. provided clinical care and contributed to clinical data collection and analyses. A. Sundararajan conducted metagenomic sequencing and data analysis and M.G. provided statistical support. M.R.S., E.G.P., and B.K.P. drafted the manuscript. All authors critically reviewed and approved the manuscript.

## Competing interests

The authors declare no competing interests.

## Additional information

[1]Department of Medicine, Section of Pulmonary and Critical Care Medicine, University of Chicago Medicine, 5841 South Maryland Ave, Chicago, IL 60637, USA. [2]Duchossois Family Institute, University of Chicago, 900 E. 57th St, Chicago, IL 60637, USA. [3]Department of Medicine, Section of Infectious Diseases & Global Health, University of Chicago Medicine, 5841 South Maryland Ave, Chicago, IL 60637, USA. [4]Department of Medicine, Section of Gastroenterology, Hepatology and Nutrition, University of Chicago Medicine, 5841 South Maryland Ave, Chicago, IL 60637, USA. [5]Department of Medicine, Section of Cardiology, University of Chicago Medicine, 5841 South Maryland Ave, Chicago, IL 60637, USA. [6]Biological Sciences Division, Biostatistics Laboratory & Research Computing Group, University of Chicago, 5841 South Maryland Ave, Chicago, IL 60637, USA. ✉e-mail: egpamer@uchicago.edu; bpatel@medicine.bsd.uchicago.edu

