## [Peer Review File · Nature Communications]

REVIEWERS' COMMENTS

Reviewer #3 (Remarks to the Author):

Dear authors,

In line with first round of reviews, I agree that the study is novel and interesting, and performed with a high technical quality. Moreover, I can agree with the authors that the study could be considered (at least in part) hypothesis-testing, however, the data would clearly benefit from additional power and experimental backup studies to be more convincing. That said, a dataset combining fecal metagenomes and microbial metabolites is unique with focus on distinguishing clinical outcome (mortality) within similar disease severity, also bearing in mind that sampling of fresh-frozen fecal material from critically ill patients could be challenging to perform within the admission period. The study could be improved by addressing the below issues.

Specific comments:

- To further underline disease similarity between the groups, addition of circulating inflammation parameters and viral load would be useful.
- I am missing abbreviations in Table 1 for SOFA and APACHE score.
- Did the authors check for differences in history of any chronic lung disease/COPD separately (outside of the Charlson index)?
- The added alpha diversity measures (Shannon etc) should also be described in the methods section.
- As agreed with previous Reviewer 1, the non-significant difference in the established short-chain fatty acids could be due to low sample size and should be mentioned as a possible limitation in the discussion section.
- A couple of recent studies reporting on gut microbiota alterations in patients with severe COVID-19 with persistent respiratory dysfunction and or other persistent symptoms are of high relevance for the topic and should preferably be cited in the introduction section (line 105) or in the discussion

(line 211) (Liu et al 2022, doi:10.1136/gutjnl-2021-325989 and Vestad et al 2022, doi: 10.1111/joim.13458)

Reviewer #4 (Remarks to the Author):

This manuscript describes the results of research study on the microbiota and a set of microbial metabolites in COVID ICU patients. Proteobacteria were over-represented in deceased patients and a set of microbially-related metabolites were associated with survival including secondary bile acids, the indole-3-carboxaldehyde and desaminotyrosine. The later known to amplify the type I interferon production which low levels are associated with COVID-19 mortality. The authors conclude from their findings that the gut-lung axis could play an important role in COVID-19 recovery.

This manuscript was very well prepared, and the findings are significant both for our understanding and treating of COVID-19 but also a field-proof that functional characterization of the microbiome-related metabolite can capture pattern that are not observed with microbiome sequencing, and thus these approaches are complementary.

As metabolomics specialist, I placed most of my attention on this topic in the manuscript. The authors profile a set of well-known microbially-related metabolites in fecal samples using either GC-MS or LC-MS.

L108 - "To address this, we profiled fecal microbiomes and metabolomes of patients admitted to the ..." On multiple occurrences in this manuscript, the authors indicated that they are profiling the fecal metabolome or looking at the "microbiome metabolite profile" or "microbiota derived". While for metagenomics is a "universal" microbe sequencing technics, this is not the case here for the metabolites monitored. As only a defined set of metabolites is analyzed with the methods employed (GC or LC-based). Such maximalist description must be reframed in such case as it is done correctly by the authors for the abstract (L59): For example, "A set of known microbially related metabolites were analyzed in the fecal samples" or being referring to their chemical class is appropriate.

L236 - 241. The authors are formulating the most significant aspect of this study. It is indeed remarkable that the monitoring of microbially related metabolites appear to be more predictive of the outcome than microbiome shotgun sequencing. And the interpretation that other host co-factor

could play a role is reasonably cautious and logical. The outcome is indeed that a rapid functional assessment would be critical in the clinics.

L257 - The difference between microbiome manipulation and augmentation isn't clear. Augmentation would be a probiotic supplementation ? But wouldn't that be manipulation ?

L641 and L648 and supplementary text - While this is claim in the manuscript The manuscript mass spectrometry data are not accessible on Metabolights (<https://www.ebi.ac.uk/metabolights/MTBLS5288>). I presumed the authors did not share it publicly yet. For reviewing purpose, these should be made public now or please find a way so I can quickly examine the mass spectrometry data (at least the qTOF ones). Also, the method description should indicate wether the LC-qTOF acquisition was made of MS1 scans only or in DDA mode, and the mass resolution and mass accuracy routinely observed for the instrument should be specified.

L727 and L752 - " and confirmed by comparison to authentic standards". Based on metabolomics standard initiative, it is critical to specify which (and how many) experimental parameters were considered and compared for the identification/annotation. (retention time/index, m/z for the main ion(s), MS/MS spectra?).

The source/provider of the "authentic standard" must be indicated in the method section.

October 13, 2022

We have modified our manuscript and have addressed the remaining issues raised by reviewers 3 and 4. Our point-by-point response follows:

Reviewer #3

In line with first round of reviews, I agree that the study is novel and interesting, and performed with a high technical quality. Moreover, I can agree with the authors that the study could be considered (at least in part) hypothesis-testing, however, the data would clearly benefit from additional power and experimental backup studies to be more convincing. That said, a dataset combining fecal metagenomes and microbial metabolites is unique with focus on distinguishing clinical outcome (mortality) within similar disease severity, also bearing in mind that sampling of fresh-frozen fecal material from critically ill patients could be challenging to perform within the admission period.

We thank the reviewer for these comments.

To further underline disease similarity between the groups, addition of circulating inflammation parameters and viral load would be useful.

We agree with the reviewer that circulating cytokine levels and viral load determinations would be informative but, unfortunately, we do not have access to serum samples from patients included in our study. Establishing correlations between microbiota-derived metabolite levels and viral loads/inflammatory cytokine levels will be the focus of future studies.

I am missing abbreviations in Table 1 for SOFA and APACHE score.

We have provided these in the legend for Table 1.

Did the authors check for differences in history of any chronic lung disease/COPD separately (outside of the Charlson index)?

We did check for associations with interstitial lung disease, COPD and asthma and did not detect any significant correlations.

The added alpha diversity measures (Shannon etc) should also be described in the methods section.

Added to methods.

As agreed with previous Reviewer 1, the non-significant difference in the established short-chain fatty acids could be due to low sample size and should be mentioned as a possible limitation in the discussion section.

We have added a paragraph outlining the limitations of our study which includes a statement that “although we have demonstrated significant associations between the progression of respiratory failure and the paucity of potentially immunomodulatory metabolites, a larger sample size might have identified additional important correlations”.

A couple of recent studies reporting on gut microbiota alterations in patients with severe COVID-19 with persistent respiratory dysfunction and or other persistent symptoms are of high relevance for the topic and should preferably be cited in the introduction section (line 105) or in the discussion (line 211) (Liu et al 2022, doi:10.1136/gutjnl-2021-325989 and Vestad et al 2022, doi: 10.1111/joim.13458)

We appreciate the reviewer’s recommendation, but both papers focus on correlations between microbiota composition and chronic or long-COVID. Neither demonstrate associations between Proteobacteria prevalence and COVID-19 associated disease. Because there is a large and growing body of literature on long-COVID and the impact of the microbiome is likely to be distinct from its impact on acute COVID-19, we do not believe these two manuscripts add relevant context to our paper.

Reviewer #4 (Remarks to the Author):

This manuscript was very well prepared, and the findings are significant both for our understanding and treating of COVID-19 but also a field-proof that functional characterization of the microbiome-related metabolite can capture pattern that are not observed with microbiome sequencing, and thus these approaches are complementary.

We thank the reviewer for these comments.

L108 - "To address this, we profiled fecal microbiomes and metabolomes of patients admitted to the ..." On multiple occurrences in this manuscript, the authors indicated that they are profiling the fecal metabolome or looking at the "microbiome metabolite profile" or "microbiota derived". While for metagenomics is a "universal" microbe sequencing technics, this is not the case here for the metabolites monitored. As only a defined set of metabolites is analyzed with the methods employed (GC or LC-based). Such maximalist description must be reframed in such case as it is done correctly by the authors for the abstract (L59): For example, "A set of known microbially related metabolites were analyzed in the fecal samples" or being referring to their chemical class is appropriate.

We understand the reviewer's point and have changed "profiled microbiomes and metabolomes" to "profiled microbiomes and targeted metabolites" on line 109.

L236 - 241. The authors are formulating the most significant aspect of this study. It is indeed remarkable that the monitoring of microbially related metabolites appear to be more predictive of the outcome than microbiome shotgun sequencing. And the interpretation that other host co-factor could play a role is reasonably cautious and logical. The outcome is indeed that a rapid functional assessment would be critical in the clinics.

We thank the reviewer for this comment.

L257 - The difference between microbiome manipulation and augmentation isn't clear. Augmentation would be a probiotic supplementation ? But wouldn't that be manipulation ?

We agree and have removed "manipulation" and have left only "augmentation".

L641 and L648 and supplementary text - While this is claim in the manuscript The manuscript mass spectrometry data are not accessible on Metabolights (<https://www.ebi.ac.uk/metabolights/MTBLS5288>). I presumed the authors did not share it publicly yet. For reviewing purpose, these should be made public now or please find a way so I can quickly examine the mass spectrometry data (at least the qTOF ones). Also, the method description should indicate whether the LC-qTOF acquisition was made of MS1 scans only or in DDA mode, and the mass resolution and mass accuracy routinely observed for the instrument should be specified. L727 and L752 - " and confirmed by comparison to authentic standards". Based on metabolomics standard initiative, it is critical to specify which (and how many) experimental parameters were considered and compared for the identification/annotation. (retention time/index, m/z for the main ion(s), MS/MS spectra?). The source/provider of the "authentic standard" must be indicated in the method section.

We have extended the supplementary methods section and now address all of the issues raised by the reviewer.